# Real-time estimation of the effective reproduction number of SARS-CoV-2 in Aotearoa New Zealand

Rachelle N. Binny[1,2,*], Audrey Lustig[1,2,*], Shaun C. Hendy[2,3], Oliver J. Maclaren[4], Kannan M. Ridings[2,3], Giorgia Vattiato[2,3,5] and Michael J. Plank[2,5]

[1] Manaaki Whenua-Landcare Research, Lincoln, New Zealand
[2] Te Pūnaha Matatini, Auckland, New Zealand
[3] Department of Physics, University of Auckland, Auckland, New Zealand
[4] Department of Engineering Science, University of Auckland, Auckland, New Zealand
[5] School of Mathematics and Statistics, University of Canterbury, Christchurch, New Zealand
[*] These authors contributed equally to this work.

Corresponding author
Rachelle N. Binny,
binnyr@landcareresearch.co.nz

## ABSTRACT

During an epidemic, real-time estimation of the effective reproduction number supports decision makers to introduce timely and effective public health measures. We estimate the time-varying effective reproduction number, $R_t$, during Aotearoa New Zealand's August 2021 outbreak of the Delta variant of SARS-CoV-2, by fitting the publicly available EpiNow2 model to New Zealand case data. While we do not explicitly model non-pharmaceutical interventions or vaccination coverage, these two factors were the leading drivers of variation in transmission in this period and we describe how changes in these factors coincided with changes in $R_t$. Alert Level 4, New Zealand's most stringent restriction setting which includes stay-at-home measures, was initially effective at reducing the median $R_t$ to 0.6 (90% CrI 0.4, 0.8) on 29 August 2021. As New Zealand eased certain restrictions and switched from an elimination strategy to a suppression strategy, $R_t$ subsequently increased to a median 1.3 (1.2, 1.4). Increasing vaccination coverage along with regional restrictions were eventually sufficient to reduce $R_t$ below 1. The outbreak peaked at an estimated 198 (172, 229) new infected cases on 10 November, after which cases declined until January 2022. We continue to update $R_t$ estimates in real time as new case data become available to inform New Zealand's ongoing pandemic response.

## INTRODUCTION

An important measure in epidemiology is the reproduction number. The basic reproduction number, $R_0$, is the average number of people infected by a single infected individual, in a fully susceptible population and in the absence of any public health measures. For the highly transmissible Delta variant of SARS-CoV-2 (the virus that causes COVID-19), $R_0$ is estimated to be at least 6, compared to around 2.5–3 for the original strain of the virus. In reality, the proportion of the population who have immunity to the virus, either through

infection or vaccination, changes during the course of an epidemic and transmission can be altered by introduction of public health measures, such as facemask use, contact tracing, quarantining infectious individuals, stay-at-home orders, and school or workplace closures. The effective reproduction number, $R_t$, is the average number of people infected by a single infected individual at a particular time $t$, accounting for this partial population immunity and changes in transmission levels due to public health measures.

Tracking estimates of the effective reproduction number in real time is useful for: (1) estimating how quickly an outbreak is likely to grow or shrink under current public health measures, to inform decisions on whether measures need to be escalated or scaled back, and (2) assessing the effectiveness of these measures for reducing transmission. Different models have been developed for estimating $R_t$ from time-series data on the number of reported cases. Each model comes with its own set of assumptions and limitations, therefore it is useful for decision makers to have a range of estimates from different models to compare. One such model, implemented in the EpiNow2 package in R statistical software (*Abbott et al., 2020a*; *Abbott et al., 2020b*; *R Core Team, 2021*), follows current best practices (*Gostic et al., 2020*) to provide estimates of the time-varying instantaneous reproduction number, case growth rate, doubling time, and case numbers by date of infection. Their modelling approach is based on a method by *Cori et al. (2013)* and *Thompson et al. (2019)*, extended to account for delays from infection to reporting.

From the start of the COVID-19 pandemic until the end of 2021, New Zealand pursued an elimination strategy (*Baker et al., 2020*). This involved a four-tier alert level system of restrictions (*Unite Against COVID-19, 2022b*), including nationwide and regional lockdown measures, which successfully eliminated community transmission of SARS-CoV-2 during outbreaks in March–May 2020, August–September 2020 and February–March 2021. On 17 August 2021, after several months with no known community transmission, New Zealand detected its first case of the Delta variant in the community, in a 58-year old man with no clear link to the border. At that time, only 38% of the population had received at least one vaccine dose, which would have been insufficient to prevent a large outbreak occurring without strong public health measures in place (*Steyn et al., 2021*; *Steyn et al., 2022*). The government therefore made the decision to move the entire country into Alert Level 4, the most stringent tier of restrictions (Table 1). Over the following days, contact tracing and surveillance testing efforts were ramped up in an attempt to ring-fence the outbreak as daily case numbers rapidly increased to 79 new cases on 26 August. Whole genome sequencing later identified possible links to infected travellers who arrived at a managed isolation and quarantine (MIQ) facility from New South Wales, Australia (*Jelley et al., 2022*).

Ten days after the introduction of Alert Level 4 restrictions, daily cases levelled off then steadily declined. Results from widespread community testing and wastewater testing suggested the outbreak was so far contained within Auckland and on 1 September regions outside of Auckland began a phased easing to Alert Level 2. Over the following weeks, daily case numbers dwindled in a long outbreak tail, with 10-20 new cases reported each day. However, elimination proved more difficult than previous outbreaks of earlier variants, particularly when transmission started to occur within unvaccinated groups in emergency

**Table 1  Dates of restrictions and key events following the August 2021 incursion of the Delta variant of SARS-CoV-2 in New Zealand.** Restriction changes occurred at 11.59pm on each date.

| Date | Restrictions and public health response |
|---|---|
| 17 August 2021 | First case of Delta variant detected in the community. Entire country moves to Alert Level 4. Contact tracing, wastewater testing and community testing ramped up. 38% of eligible population (over-12s) has had at least one vaccine dose. |
| 31 August | Regions south of Auckland to Alert Level 3. |
| 2 September | Northland to Alert Level 3. |
| 7 September | Regions outside of Auckland to Alert Level 2. |
| 21 September | Auckland and Upper Hauraki to Alert Level 3. |
| 25 September | Upper Hauraki to Alert Level 2. |
| 3 October | Two cases detected in Raglan and Hamilton. Parts of Waikato move to Alert Level 3. |
| 5 October | Roadmap step 1 introduced in Auckland, allowing up to 10 people from two households to gather outdoors and reopening of Early Childhood Education centres. Transition from elimination strategy to suppression strategy. |
| 7 October | Waikato Alert Level 3 boundary extended to include additional districts. |
| 8 October | Two positive cases are discovered to have travelled out of Auckland into Northland. Northland moves to Alert Level 3. |
| 16 October | 'Super Saturday' vaccination drive: 85% of eligible population (aged 12 and over) have had at least one vaccine dose, 66% have received two doses. |
| 19 October | Northland to Alert Level 2. |
| 27 October | Waikato to Step 1 of Alert Level 3. 87% of eligible population have had at least one vaccine dose, 72% have received two doses. |
| 2 November | Upper Northland to Alert Level 3 after detection of two community cases with no known epidemiological links. Waikato to Step 2 of Alert Level 2, allowing up to 25 people from multiple households to meet outdoors. Retail stores and public venues including libraries, museums and zoos can open. Hospitality venues and close contact businesses remain closed. |
| 9 November | Auckland to Step 2 of Alert Level 3. |
| 10 November | 89% of eligible population have had at least one vaccine dose, 79% have received two doses. |
| 11 November | Upper Northland to Alert Level 2. |
| 16 November | Waikato to Alert Level 2. |
| 2 December | Introduction of 3-tier COVID-19 Protection Framework (traffic lights). Following regions at Red: Northland, Auckland, Taupō and Rotorua Lakes Districts, Kawerau, Whakatane, Ōpōtiki Districts, Gisborne District, Wairoa District, Rangitikei, Whanganui and Ruapehu Districts. All other regions at Orange. 93% of eligible population have had at least one vaccine dose, 86% have received two doses. |
| 14 December | Auckland travel boundary lifted. |
| 15 December | 94% of eligible population have had at least one vaccine dose, 90% have received two doses. |
| 29 December | First border-related Omicron case with community exposures. |
| 30 December | All regions to Orange except Northland. |
| 20 January 2022 | Northland to Orange. |
| 23 January | Government announces detection of nine community cases with the Omicron variant and multiple community exposure events. |

and transitional housing, where public health measures including contact tracing and self-isolation were more challenging or impossible. From 22 September, restrictions in Auckland were eased to Alert Level 3 and, following detections in the neighbouring regions of Waikato and Northland, these regions moved up to Alert Level 3.

On 6 October, despite a rise in case numbers, the government announced further easing of some restrictions for Auckland and case numbers continued to steadily grow. This decision marked a transition from New Zealand's elimination strategy to one of suppression: using restrictions to control case numbers to low levels while continuing to increase vaccination coverage in the population through an accelerated vaccine roll-out. Modelling suggested that well over 90% of New Zealand's eligible population, aged 12 years and over, need to be fully vaccinated (two vaccine doses) to safely control community transmission without the need for strong public health measures, such as stay-at-home orders and workplace closures (*Steyn, Plank & Hendy, 2021*; *Steyn et al., 2022*). By mid-November, with close to 80% of the eligible population vaccinated with two doses and all regions except Auckland at Alert Level 2, the Delta outbreak reached its peak at approximately 200–220 new cases per day. Cases steadily declined from this time until January 2022 and by 18 January there were a total of 11,396 reported community cases and 26 deaths related to the Delta outbreak. The four-tier Alert Level system was replaced with a 3-tier COVID-19 Protection Framework (*Unite Against COVID-19, 2022a*) (traffic light levels, with red being the highest restriction setting, focused on minimising spread and providing protection through high vaccination coverage) which came into effect on 2 December 2021. Following the detection of nine community cases infected with the highly transmissible Omicron variant announced on 23 January, cases numbers rapidly increased again and New Zealand experienced its largest COVID-19 outbreak to date, peaking in mid-March 2022.

The ability to track changes in case numbers and the reproduction number in real time is critical to inform decisions about when and where additional measures should be introduced to prevent high case numbers overwhelming healthcare and contact tracing systems, and when these measures can be safely eased. In this work, we use EpiNow2 to estimate the time-varying effective reproduction number $R_t$ in New Zealand during the August 2021–January 2022 outbreak of the Delta variant. The EpiForecasts team who developed EpiNow2, apply the model to global, publicly available case data, aggregated using the covidregionaldata R package (*Abbott et al., 2020a*; *Abbott et al., 2020b*; *Abbott et al., 2020c*; *R Core Team, 2021*), and publish real-time estimates of the effective reproduction number on their webpage (https://epiforecasts.io/covid/) (*Abbott et al., 2020b*). However, there is value in applying this method using more detailed line list data held by Ministry of Health, which allows model parameters to be tailored specifically to the New Zealand setting. We compare our estimates to these EpiForecasts estimates and summarise the differences in assumptions made in our implementation of the model. Throughout New Zealand's Delta outbreak and in the current Omicron outbreak, we continue to update our estimates regularly as new data become available and provide these to Ministry of Health to inform operational planning and decision-making in real time.

## MATERIALS & METHODS

### Data

Data on daily numbers of COVID-19 cases in New Zealand's Delta outbreak, extracted from the EpiSurv database (administered by ESR), were obtained from Ministry of

Health (MOH). Data fields included the date of reporting, date of symptom onset (where available), whether the case was internationally-imported (yes, no or unknown), and status ('confirmed', 'probable' or 'under investigation'). Only confirmed and probable cases were included in our analyses. The EpiSurv database is updated in real time and cases 'under investigation' generally arise in the last 1–3 days of each instance of the database; in subsequent updates, they are either removed or updated to 'confirmed' or 'probable'. Exclusion of these cases may introduce a downward bias in effective reproduction number estimates at the end of the time-series. However, cases under investigation may include some individuals who ultimately test negative and reporting practices may vary consistently between different District Health Boards so these data cannot be reliably included.

Here, we analyse data extracted from the EpiSurv database on 20 January 2022 on all cases ($n = 12{,}460$; 12,441 confirmed, 19 probable, 0 under investigation) reported between 17 August 2021 and 19 January 2022. Case data on the last day of the time-series ($n = 50$) are likely right-truncated due to an unrecorded delay between a case returning a positive test result and being recorded in the dataset. For input to the model, we therefore truncated the time series to exclude the last day of data. We also excluded the internationally-imported cases ($n = 1{,}014$), all of which were detected in MIQ and had no exposure to the community. This left 11,378 confirmed cases and 18 probable cases in the line list which were aggregated to give the daily totals of new reported cases.

## Model

We implement the EpiNow2 package *Abbott et al. (2020a)* to estimate a time-varying effective reproduction number, $R_t$, during New Zealand's Delta outbreak, using data on daily reported case numbers from 17 August 2021 to 18 January 2022. A detailed description of the model is given in *Abbott et al. (2020b)*. EpiNow2 implements a Bayesian latent variable approach, with the following key modelling steps:

1. Starting with an estimate of the initial number of infections (with a prior based on initial case numbers), incidence is projected forward in time on a daily time step. Here, incidence is defined as the number of individuals who are infected at time $t$, and who will eventually be tested and reported as a case. We do not model undetected infections, such as asymptomatic infections, that do not appear in case data. The incidence, $I_t$, at time $t$ is estimated by summing over the imputed incidences $I_{t-\tau}$ from previous time steps $\tau$, weighted by the generation time distribution, and multiplying by the estimate of $R_t$ at time $t$.

2. Temporal variation in $R_t$ is accounted for using an approximate Gaussian process: $R_t \sim R_{t-1} \times$ (squared exponential kernel).

3. These incidences $I_t$ are mapped to the mean reported cases, $D_t$, at time $t$ by convolving over the incubation period distribution and the onset-to-reporting delay distribution.

4. The observed number of reported cases, $C_t$, at time $t$ is assumed to be negative-binomially distributed with overdispersion $\phi$ and mean ($D_t \omega_{d,t}$), where $\omega_{d,t}$ is a day of the week effect (*i.e.*, seven independent parameters) that scales the mean reported cases at time $t$ according to the day of the week.
Following *Abbott et al. (2020b)*, we employ a gamma-distributed generation time distribution sourced from *Ganyani et al. (2020)* but refit using the log-normal incubation period distribution reported by *Lauer et al. (2020)* with mean 5.2 (SD 1.1) days and SD of 1.52 (SD 1.1) days. This results in a mean generation time of 3.6 (SD 0.7) days and standard deviation 3.1 (SD 0.8) days. Both *Ganyani et al. (2020)* and *Lauer et al. (2020)* used data on the original strain of SARS-CoV-2. An analysis of household transmission data by *Hart et al. (2022)* estimated a mean intrinsic generation time for the Delta variant of 4.7 (95% CrI 4.1, 5.6) days and a mean household generation time of 3.2 (2.5, 4.2) days, which were lower than estimates for the Alpha variant (5.5 [4.7, 6.5] days and 4.5 [3.7, 5.4] days for intrinsic and household generation times respectively). We refit the model using these shorter and longer mean generation times to assess to what extent it affected our results.

The delay from symptom onset to reporting is estimated by fitting a log-normal distribution, using Stan (*Stan Development Team, 2021*), to 100 subsampled bootstraps of the onset-to-report delays from 3208 cases reported between 17 August 2021 and 18 January 2022 (excluding nine cases with delays greater than 60 days), taking (with replacement) 250 samples from these delays in each bootstrap. The distribution is adjusted to account for left and right censoring in the data due to dates being rounded to the nearest day. For computational reasons, the distribution is truncated to the maximum observed delay. In preliminary analyses, we checked for temporal variation in reporting delays by performing independent fits of a log-normal distribution to the delays recorded in each month. There was little change in distribution parameters between different months (Table S2) so we assumed the delay distribution was static over the full time period. There was insufficient data to assess whether delays varied spatially. The model was fitted to the time-series of new reported cases per day using Markov-chain Monte Carlo (MCMC), implemented in Stan (*Stan Development Team, 2021*). We used four chains with a warmup of 500 steps each and 4,000 samples post-warmup. Convergence was assessed using the Rhat diagnostic.

The model outputs the posterior estimates of the inferred incidence $I_t$ (*i.e.*, new cases at their time of infection), the mean reported cases $D_t \omega_{d,t}$, (*i.e.*, new cases at their time of reporting, having accounted for incubation periods, reporting delays, and day of the week effects) and the instantaneous effective reproduction number $R_t$ over the modelled time period. The latter is defined as the expected number of new secondary infections per infectious individual at time $t$, scaled by their relative infectiousness at time $t$ (*Gostic et al., 2020*). Estimates over the last nine days of the time-series are based on partial data because the delays from infection to reporting mean there are very likely some cases who were infected on these dates but have yet to appear in reported case data. The time-varying growth rate, $r_t$, is also estimated from the time-varying effective reproduction number using an approximation derived by *Park et al. (2019)*. Doubling time (or halving time when growth rate is negative) is calculated as $\ln(2)/r_t$. In Results, we provide figures showing the estimated 20%, 50% and 90% credible intervals (CrI) of the posterior distributions for $(D_t \omega_{d,t})$, $I_t$, $R_t$ and $r_t$. We do not explicitly model non-pharmaceutical interventions or vaccination so cannot directly quantify their effects on transmission. However, given that levels of immunity from prior infections were very low during the period studied, these two factors were the leading drivers of variation in transmission in this period so we discuss
how changes in $R_t$ coincided with changes in interventions and increasing vaccination coverage. The EpiNow2 package allows forecasting of estimates over a 14-day time horizon but we do not implement that functionality here. Model assumptions and how they differ from the implementation by EpiForecasts, used to generate the New Zealand estimates published on their webpage (https://epiforecasts.io/covid/posts/national/new-zealand/), are detailed in Table 2.

## RESULTS

Using subsampled bootstraps of the delays from symptom onset to reporting, up to a maximum of 34 days, we fitted a lognormal distribution of onset-to-report delays with a mean of 3.84 days and standard deviation of 3.94 days (parameters: $\mu = 0.988$ (SD $= 0.088$) and $\sigma = 0.847$ (SD $= 0.063$), which was a good visual match to the data (Fig. S1). Fitting the EpiNow2 model to the 22-week time-series of daily reported case numbers, we obtained estimates for the time-varying mean reported cases, incidence, effective reproduction number and growth rate of New Zealand's Delta outbreak up to 18 January 2022. The Rhat statistic was less than 1.05 for all parameter estimates indicating model convergence. The estimated daily number of new cases (ribbons) are shown alongside the actual reported case numbers (grey bars) in Fig. 1. Mean reported case counts ($D_t \, \omega_{d,t}$) by date of report (Fig. 1A), were a good visual match to the data, first peaking at a median 57 (90% CrI 32, 90) cases on 27 August 2021 (cf. 79 on 26 August in actual reported cases). Plotting cases by their estimated date of infection ($I_t$) shows the first peak in incidence occurred on 20 August with a median 73 (48, 101) new cases infected that day (Fig. 1B).

Figure 2 shows the change in estimated effective reproduction number ($R_t$) over time alongside key timings of changes to levels of restrictions. On 17 August, $R_t$ is greater than one but immediately starts to decline after introduction of Alert Level 4 restrictions from 18 August. This, along with the decline in incidence from 20 August, suggests that these restrictions had a near-immediate effect on reducing transmission, though this reduction did not filter through to reported case numbers for another 7–10 days. After twelve days under Alert Level 4 restrictions, the median $R_t$ was reduced to 0.57 (0.42, 0.76), growth rate $r_t$ was −0.13 (−0.18, −0.07) per day and doubling time was −5.48 (−3.89, −10.26) days (Figs. 2–3). Estimated daily cases by infection date were reduced to a minimum of 14 (11, 18) per day by 6 September (Fig. 1B) and the outbreak entered its long tail. After this time, the reproduction number and daily infections steadily increased, with $R_t$ growing to values greater than one from 11 September. Coinciding with the further easing of some restrictions in Auckland during September and October, median $R_t$ remained over 1 and New Zealand experienced its largest outbreak since the start of the pandemic. Transmission was at its highest on 2 October, with median $R_t = 1.27$ (1.15, 1.38), corresponding to a growth rate of 0.07 (0.04, 0.10) per day and doubling time of 9.66 (16.68, 6.86) days. However, by mid-November, regional Alert Level 3 restrictions remained in place and vaccination coverage had rapidly increased from 38% of the eligible population having received at least one vaccine dose on 17 August to 89% by 10 November. The median $R_t$ subsequently decreased below 1, causing the outbreak to peak at an estimated 198 (172,

Binny et al. (2022), *PeerJ*, DOI 10.7717/peerj.14119

**Table 2  Model assumptions (those that differ from EpiForecasts in bold text).**

| Parameter/dataset | Our implementation | EpiForecasts |
|---|---|---|
| Data source | **Ministry of Health line list data from EpiSurv database, administered by ESR. Confirmed and probable cases, excluding internationally-imported cases in MIQ and cases under investigation.** | World Health Organisation (2020) data on confirmed cases, including internationally-imported cases in MIQ. |
| Model fitting | Four chains with a warmup of 500 each and 4,000 samples post-warmup. | |
| Symptom onset-to-reporting delay distriution | Log-normal distribution with **mean of 3.84 days and standard deviation of 3.94 days [parameters: $\mu = 0.988$ (SD $= 0.088$) and $\sigma = 0.847$ (SD $= 0.063$)]. Maximum delay is 34 days**. Fitted to delays in case data from 17 August 2021 to 18 January 2022; Fig. S1. | Log-normal distribution with mean of 6.5 days and standard deviation of 17 days. Maximum delay is 30 days. Fitted to combined data from every country for which onset-to-report delays are available in a publicly accessible linelist (*Abbott et al., 2020a*; *Abbott et al., 2020b*; *Kraemer et al., 2020*). |
| Incubation period distribution | Log-normal with mean of 5.2 days (SD 1.1) and SD of 1.52 days (SD 1.1) (*Lauer et al., 2020*) | |
| Generation time distribution | *Main analysis:* Gamma distribution with a mean of 3.64 days (SD 0.71) and SD of 3.07 days (SD 0.77), sourced from *Ganyani et al. (2020)* but re-fit using *Lauer et al. (2020)* incubation period. *Shorter generation time:* mean 3.2 (SD 0.46) days and SD 2.4 (SD 0.33) days (*Hart et al., 2022*). *Longer generation time:* mean 4.6 (SD 0.4) days and SD 3.1 (SD 0.2) days (*Hart et al., 2022*). | |
| Prior on $R_t$ at $t = 0$ | Log-normal with mean and standard deviation of 1. | |
| Gaussian process kernel | Squared exponential kernel. Length scale was given a log-normal prior with a mean of 21 days and standard deviation of 7 days truncated to be greater than 3 days and less than the length of the data. The prior on the magnitude was standard normal. | |

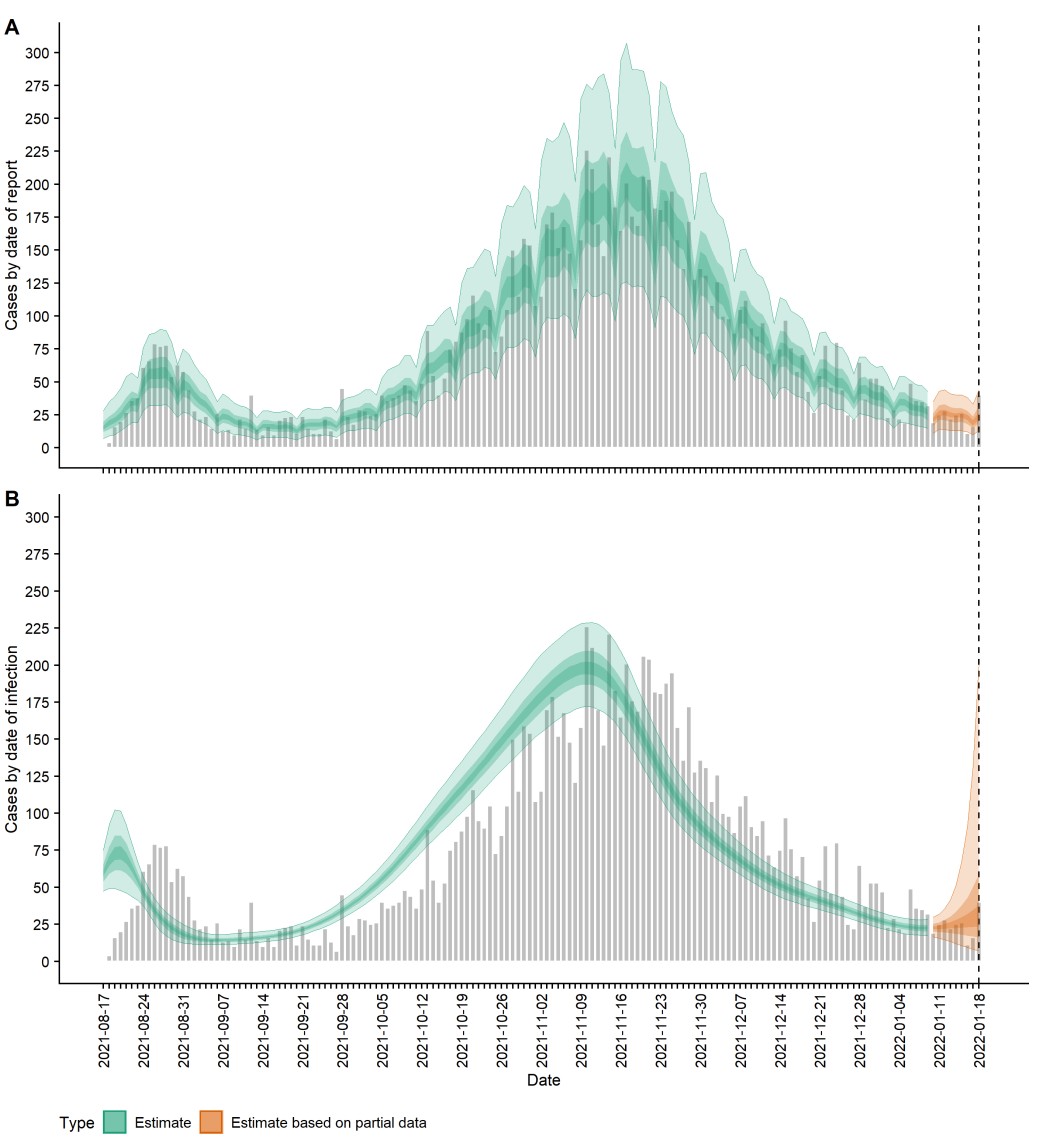

**Figure 1 COVID-19 cases (observed and estimated) over time.** (A) Observed cases (confirmed and probable) by date of report (grey bars) and estimated mean reported cases, $D_t \omega_{d,t}$, by date of report (ribbon); (B) Observed cases by date of report (grey bars) and estimated incidence $I_t$ (cases by date of infection, having accounted for delays from infection to reporting) (ribbon). Lightest ribbon = 90% credible interval (CrI); darker ribbon = 50% CrI; darkest ribbon = 20% CrI. Estimates up to 9 January 2022 are based on full data (green); estimates from 10 January to 18 January (orange) are based on partial data and have been adjusted for right truncation of infections.

229) new cases infected on 10 November (cf. 226 cases reported on 10 November in actual data). From this point, median $R_t$ remained relatively constant at around 0.8 to 0.9 as cases continued to decline and the COVID-19 Protection Framework was introduced. By January 2022, prior to detection of community cases with the Omicron variant, the outbreak had decreased to a median 20–30 cases infected per day.

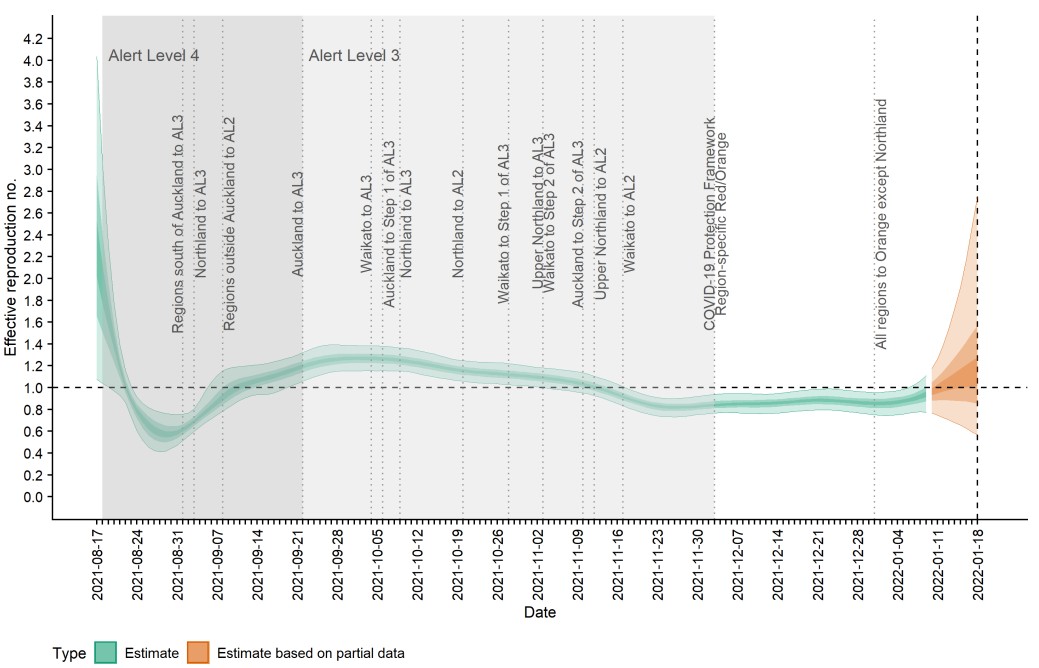

**Figure 2** **Estimated effective reproduction number, $R_t$, over time and timings of interventions.** Lightest ribbon = 90% credible interval (CrI); darker ribbon = 50% CrI; darkest ribbon = 20% CrI. Estimates up to 9 January 2022 are based on full data (green); estimates from 10 January to 18 January (orange) are based on partial data and have been adjusted for right truncation of infections.

Note that during the first 6.5 weeks of the outbreak, the vast majority of reported cases were detected only in the Auckland region. On 3 October, cases were detected in Waikato and from this time on transmission occurred within both regions but likely not between the two regions due to the Auckland travel boundary in place until 14 December. This is supported by whole genome sequencing which suggested that the Waikato cluster arose from a single introduction from Auckland (*Hadfield et al., 2018*; *Khare et al., 2021*), while clusters detected in Northland in late October—early November likely represented multiple introductions (*New Zealand Ministry of Health, 2021*).

Refitting the model using the mean generation times of *Hart et al. (2022)* resulted in similar estimates of $R_t$ as shown in Fig. S3. With the shorter mean generation time of 3.2 days, $R_t$ reached its minimum at a median 0.73 (0.53, 0.83) on 30 August and its maximum of 1.24 (1.15, 1.32) on 1 October. Assuming a longer mean generation time of 4.6 days resulted in a minimum $R_t$ of 0.52 (0.40, 0.72) on 30 August and maximum on 1.32 (1.18, 1.46) on 5 October. Summary statistics for all posterior estimates are provided in Table S1.

Estimates of daily numbers of reported cases by report date and infection date, and the effective reproduction number published by EpiForecasts are shown in Fig. S2, for two 16-week periods up to 15 October 2021 and up to 28 January 2022. The EpiForecasts estimates were broadly similar to our estimates but tended to over-estimate the numbers of new infections each day, particularly in periods of low local transmission. Epiforecasts fit to data published by the World Health Organisation (*World Health Organisation, 2020*),

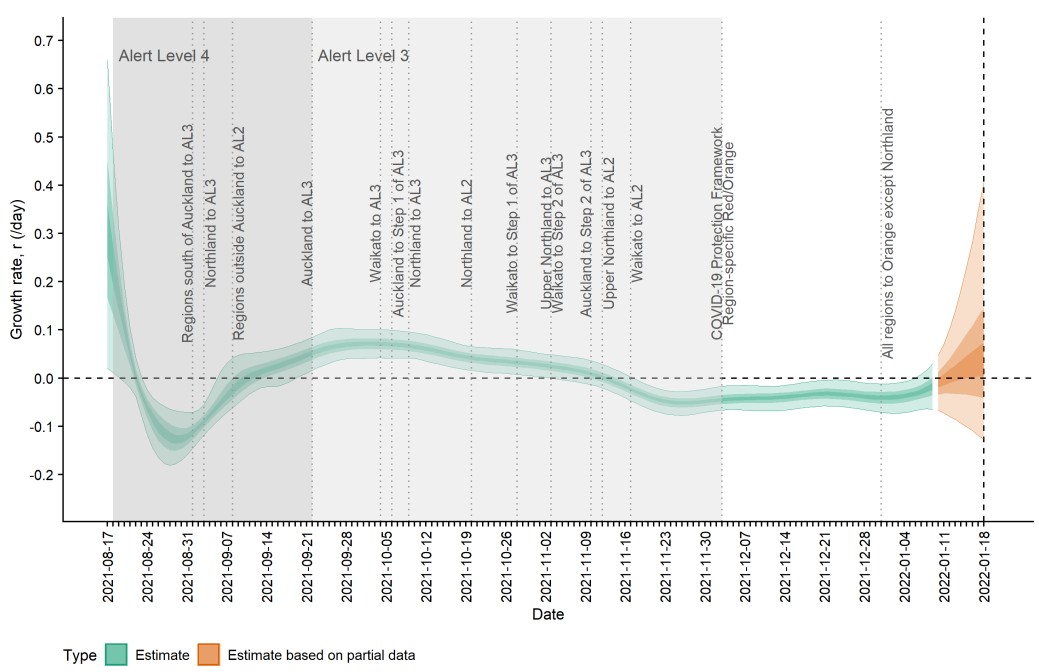

**Figure 3** **Estimated rate of exponential growth, $r_t$, over time and timings of interventions.** Lightest ribbon = 90% credible interval (CrI); darker ribbon = 50% CrI; darkest ribbon = 20% CrI. Estimates up to 9 January 2022 are based on full data (green); estimates from 10 January to 18 January (orange) are based on partial data and have been adjusted for right truncation of infections. Doubling time is related to growth rate by $\ln(2)/r_t$.

which sources data from the New Zealand Ministry of Health (*New Zealand Ministry of Health, 2022*), but does not include information on the importation status of New Zealand's cases. Case numbers modelled by EpiForecasts were therefore biased upwards by inclusion of internationally-imported cases detected in MIQ facilities. As the EpiForecasts model attributes all new imported cases to local transmission, this could result in over- or under-estimation of $R_t$. We were able to isolate the domestically-acquired cases for model fitting to provide less biased estimates of community transmission occurring in New Zealand.

## DISCUSSION

Our study demonstrates how real-time estimation of the effective reproduction number can effectively support a timely public health response during a community outbreak. New Zealand's elimination strategy and use of stringent stay-at-home measures were successful at controlling several outbreaks of earlier variants even while vaccination coverage remained low. However, they were less effective at controlling the highly transmissible Delta variant and an accelerated vaccination rollout in combination with ongoing regional restrictions were ultimately required to reduce $R_t$ below one. For community outbreaks where a significant proportion of cases are internationally-imported, models should distinguish between imported and domestic infections to avoid biasing estimates of transmission.

Delays from infection to reporting mean that cases reported on a particular day are really indicative of the transmissibility approximately 7–10 days prior. The EpiNow2 framework accounts for these delays, allowing case counts to be mapped from their date of reporting to date of infection to estimate an instantaneous measure of transmissibility under current conditions. While there can be larger uncertainty around $R_t$ estimates closer to 'now' (due to these estimates being based on less data), the EpiNow2 approach is particularly well-suited for informing real-time decisions and facilitating a rapid public health response to changes in transmission. It also allows observed changes in $R_t$ to be related to changes in interventions or other variables affecting transmission at certain times. More naive approaches that estimate the reproduction number directly from reported cases without accounting (or only partially accounting) for these delays provide a lagged view of an outbreak's trajectory and may obscure the temporal relationship between changes in policy or behaviour and transmission dynamics (*Gostic et al., 2020*).

The instantaneous reproduction number $R_t$ estimated by EpiNow2 is a property of the epidemic at a particular time $t$. It measures the average number of secondary infections per infectious individual at time $t$, assuming that the current conditions remain unchanged until those individuals who were infectious at time $t$ have recovered. This differs from another widely computed measure, the case reproduction number $R_c$, which is the average number of people an individual infected at time $t$ eventually infects, with no assumption of conditions being unchanged after time $t$. $R_c$ is the measure estimated by Wallinga and Teunis-type methods, though *Wallinga & Teunis (2004)* consider individuals with symptom onset at time $t$ as opposed to infection at $t$ (*Cori et al., 2013*). In a situation where a change in conditions (such as the introduction of stay-at-home measures) brings about an abrupt change in transmissibility, the instantaneous reproduction number $R_t$ will also change abruptly at that time. In contrast, $R_c$ will transition smoothly because it estimates the number of people each case will actually infect, accounting for the fact that the infectious periods of cases infected prior to the change in conditions may span times before and after this change in transmission. Estimates of $R_c$ rely on information about future incidence so can only be obtained in retrospect, and suffer from right censoring of reported case data in an epidemic that is still in progress. For estimates of the instantaneous $R_t$, which are calculated from current and past incidence, right censoring causes fewer issues, especially where the delays to reporting are accounted for; this measure is therefore more appropriate for tracking transmissibility in real-time (*Fraser, 2007*).

There are several important limitations of the EpiNow approach. First, the model assumes that the proportion of infections that are detected remains constant over the study period, *i.e.*, consistent methods and effort for surveillance (*e.g.*, testing, contact tracing) and following consistent case definitions. This is assumption is likely violated; in reality, the level of under-reporting varies over time as contact tracing efforts are ramped up and surveillance testing rates vary over time and over different regions. This was certainly the case over the first few weeks of the outbreak. However, such changes will only bias estimates temporarily if surveillance becomes consistent again after a change. The period of time required before estimates become unbiased depends on the generation time distribution,

the delay distributions, and the correlation length scale of the Gaussian process used to model temporal variation in $R_t$.

Model estimates are sensitive to the choice of distribution for the delay from symptom onset to reporting. This distribution affects the upscaling of cases by date of onset that accounts for cases that have onset but have yet to be reported. We used a distribution estimated by fitting to data from New Zealand's Delta outbreak. Reporting delays can vary over the course of an outbreak or between regions, especially early on in an outbreak or during periods of high transmission where the testing and contact tracing system may come under significant strain. The model assumes this distribution is static over time, which was a reasonable assumption for this outbreak (Table S2) but may not hold for future outbreaks in New Zealand or in other countries. If the true delay from onset to reporting at a given time is shorter than the estimated static delay distribution, then the model will overestimate onset case numbers, and vice versa for true delays longer than the distribution used. The bootstrapped subsampling approach somewhat mitigates these issues by allowing multiple delay distributions based on the observed data to be considered at the cost of increasing uncertainty in the estimates. However, if significant temporal or spatial variation in reporting delays is suspected in future outbreaks then using other estimation procedures that are robust to such changes (see *e.g., Li & White, 2021*; *Günther et al., 2021*) may be more appropriate.

Using distributions of the incubation period and onset-to-reporting delay to infer incidence from observed reported cases tends to result in over-smoothing of the incidence. The assumed distributions for generation time and incubation period sourced from *Ganyani et al. (2020)* and *Lauer et al. (2020)* were estimated by fitting to data on infections with the original strain of the virus in other countries and these distributions may differ for the Delta variant. However, the mean generation time used here sits between two estimates for Delta reported by *Hart et al. (2022)* and re-ftting the model using these alternative mean generation times had little impact on our estimates of $R_t$. Furthermore, the model incorporates uncertainty in the means and standard deviations of both distributions (as well as in the reporting delay distribution), which goes some way to accounting for variation from the expected distributions, for example due to different populations or variants. For analyses of future outbreaks, parameters for the incubation period, generation time and reporting delay distributions should continue to be calibrated using up-to-date data as it becomes available. In particular, recent studies have suggested the mean generation time for the Omicron variant may be shorter than the Delta variant (*Abbott et al., 2022*).

As with most models, EpiNow2 estimates are more reliable when daily case numbers are large and the time series contains at least 14 days of non-zero cases. When analysing small outbreaks, slight changes in daily case numbers can lead to large variations in model estimates. Such estimates should be treated with caution and interpreted alongside other information sources, for example the detailed case-specific information collected by contact tracers. Case numbers were too low to robustly estimate regional trends during New Zealand's Delta outbreak using this technique. We were able to obtain reliable estimates at national scale, though the estimates based on partial data (orange bands in Figures) in the last 9 days of the time-series are relatively unstable and sensitive to the number of reported

cases on the final time point. These values may change significantly as data are updated and new time points are added to the series.

We do not explicitly model interventions or vaccination so could not establish their relative contributions for reducing transmission. Other models attempting to infer the relative effectiveness of interventions and vaccination (see *e.g.*, *Li et al., 2022*) generally require data from multiple countries that have implemented similar measures and with varying levels of vaccination coverage. As well as changes to levels of restrictions and vaccination, other factors can also make it challenging to attribute a change in $R_t$ to a particular public health measure. For example, variability in the level of compliance with public health measures over time (*e.g.*, willingness to wear facemasks, adherence to restrictions on between-household contact) and between different subgroups in the population influences $R_t$. Transmission levels will also vary between subgroups of the population, due to different demographics, levels of vaccination, and different environmental conditions, such as average household size and density of housing. The model masks this heterogeneity so the estimates of the time-varying $R_t$ are an attribute of the outbreak, in whichever subgroups it is occurring at that particular time. This has important implications for small outbreaks, where the subgroups in which the virus is spreading may not be representative of the wider population. If case numbers are high enough for reliable inference, such fragmented outbreaks may be better modelled as separate local outbreaks, for example regional-level models for different district health boards (*Vegvari et al., 2021*). In New Zealand's Delta outbreak, different sub-clusters have been identified (*Jelley et al., 2022*) but there were insufficient case numbers to fit the model for each sub-cluster separately. It is therefore not possible to disentangle the relative contributions that public health measures and compliance levels had on transmission, from other attributes of the sub-clusters dominating the outbreak at that particular point in time. For example, the increase in transmission towards the end of September could have been driven by the easing of restrictions in Auckland, changing levels of public compliance, the virus spreading into under-vaccinated groups or populations staying in emergency and transitional housing, or the combination of these factors.

## CONCLUSIONS

Our analysis of New Zealand case data demonstrates the effectiveness of high vaccination coverage in combination with public health measures for controlling an outbreak of the Delta variant of SARS-CoV-2. Modelling approaches that account for delays from infection to reporting reveal important changes in underlying transmission dynamics that may not be apparent when dealing only with reported cases. Real-time estimation of the effective reproduction number remains critical for informing decision-making and operational planning in New Zealand's ongoing pandemic response.

## ACKNOWLEDGEMENTS

The authors acknowledge the support of the New Zealand Ministry of Health and the Institute of Environmental Science and Research in supplying data for these analyses.

Results were generated using the open-source EpiNow2 model developed by S. Abbott, J. Hellewell, K. Sherratt, et al. (2020), and made use of the estimates and approach of S. Abbott, J. Hellewell, R. N. Thompson, et al. (2020). We acknowledge the use of New Zealand eScience Infrastructure (NeSI) high performance computing facilities and consulting support as part of this research.

### Funding

This work was funded by the New Zealand Government. New Zealand's national high performance computing facilities are provided by NeSI and funded jointly by NeSI's collaborator institutions and through the Ministry of Business, Innovation & Employment's Research Infrastructure Programme (https://www.nesi.org.nz). The data for this work were provided by New Zealand Ministry of Health and the Institute of Environmental Science and Research.

### Grant Disclosures

The following grant information was disclosed by the authors:
New Zealand Government.
Ministry of Business, Innovation & Employment's Research Infrastructure Programme.

### Competing Interests

This work was funded by the New Zealand Government.

### Author Contributions

- Rachelle N. Binny conceived and designed the experiments, performed the experiments, analyzed the data, prepared figures and/or tables, authored or reviewed drafts of the article, and approved the final draft.
- Audrey Lustig conceived and designed the experiments, performed the experiments, analyzed the data, prepared figures and/or tables, authored or reviewed drafts of the article, and approved the final draft.
- Shaun C. Hendy conceived and designed the experiments, authored or reviewed drafts of the article, and approved the final draft.
- Oliver J. Maclaren conceived and designed the experiments, authored or reviewed drafts of the article, and approved the final draft.
- Kannan M. Ridings conceived and designed the experiments, authored or reviewed drafts of the article, and approved the final draft.
- Giorgia Vattiato conceived and designed the experiments, authored or reviewed drafts of the article, and approved the final draft.
- Michael J. Plank conceived and designed the experiments, analyzed the data, authored or reviewed drafts of the article, and approved the final draft.

## Data Availability

The data on daily case numbers used in the analysis are available in the Supplementary File.

## Supplemental Information

Supplemental information for this article can be found online at http://dx.doi.org/10.7717/peerj.14119#supplemental-information.

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
