# Peer review of "Real-time estimation of the effective reproduction number of SARS-CoV-2 in Aotearoa New Zealand"

_PeerJ, doi:10.7717/peerj.14119_

## Round 0.1 · original submission · Major Revisions

Three reviewers kindly offered comments for improvement. I agree with one of the reviewers that comparing estimated Rt against R0 of delta variant is very tricky one. Also, accounting for reporting delay over the course of time is an important subject to be addressed. Please consider conducting a reanalysis to address the point.

Reviewer 1 ·

Basic reporting

The authors estimated the time-varying effective reproduction number in New Zealand and showed the impact of increases in the Alert Level on the transmission dynamics of COVID-19. The manuscript is well written. However, there are some aspects of the manuscript that could be made clearer and I hope this suggestion makes this manuscript have the highest impact.

Experimental design

The authors stated that vaccination coverage with a single dose was only 38%, prior to the declaration of Alert Level 4. However, I believe that vaccination coverage has improved over time, and not only the declaration of Alert Level 4, but also the nationwide vaccination caused the decrease in the effective reproduction number. Thus, I would incline that such changes in the vaccine coverage should be taken into account (in the renewal process) when the goalof estimating the effective reproduction number is to assess the impact of NPIs on the transmission dynamics, as is the case in this study (not to monitor the transmission dynamics).

Validity of the findings

Since various NPIs have been continuously implemented and risk awareness has been raised, I am not convinced that comparing the estimated effective reproduction number to the assumed basic reproduction number of the Delta variant (which is 6) is optimal. So I propose comparing those estimates to the previously estimated effective reproduction number during the Alert Level 3 period. However, because of the increased vaccine coverage and potential effects on transmissions, the above comment should be applied here as well.
In addition, I'm curious as to why the modeled result in Figure 1B is not well-matched with the case counts by date of infection (which the R package provided).

Additional comments

I believe that the current manuscript may not be suitable for international readers because it primarily demonstrated time-varying reproduction numbers in New Zealand using the R package. Therefore, I would suggest that the authors state the main messages or practical implications that this study can provide to international readers in the Discussion section.

Reviewer 2 ·

Basic reporting

Good.

Experimental design

Good.

Validity of the findings

Good.

Additional comments

I enjoyed reading this paper, and I think it is well-written in general. I have a couple of comments below:

1-I think the result section should be expanded, by particularly moving some parts of the discussion to the result section and adding some additional results (via sensitivity analyses). Specifically, the authors should check how Omicron variant and its shortened generation interval (incubation period) affect their results, ideally through some new sets of parameters in EpiNow2. Also, the authors mentioned the impact of data availability in their discussion. I would like to know if the New Zealand data is good enough for reliable estimation.

2. For the figures: I am not sure I understand the meaning of “Crl”, “90% Crl”, “50% Crl” and so on. My guess is that it refers to credible interval, if this is the case why include 90%, 50% and 20% CI?

3. I got lost when the authors discussed the impact of vaccination. How did the authors incorporation vaccination coverage (which is time-varying) into the model?

4. Modeling reporting delays is a key here, and the authors also mentioned that the assumption that the reporting delay distribution is static was likely violated. How did the authors deal with the scenario when the reporting delay distribution is not constant and likely varies at different stages and regions? I am not convinced that the bootstrap method is robust enough. There are other approaches could be potentially helpful here (for example, Li & White (2021)). Please include a literature review on reporting delay adjustment and preferably double check your results using a different approach.

Reference:

Li T, White LF. (2021). Bayesian back-calculation and nowcasting for line list data during the COVID-19 pandemic. PLOS Computational Biology 17(7): e1009210. https://doi.org/10.1371/journal.pcbi.1009210

Reviewer 3 ·

Basic reporting

N/A

Experimental design

N/A

Validity of the findings

N/A

Additional comments

See attached PDF

Annotated reviews are not available for download in order to protect the identity of reviewers who chose to remain anonymous.

---

## Round 0.2 · Minor Revisions

Please address remaining comments.

Reviewer 1 ·

Basic reporting

Overall the manuscript is well revised.

Experimental design

Due to the rapid change in the vaccination coverage (38% to 89% within 3 months), I am still not fully convinced about discussing the effect of interventions without integrating it into the Rt estimation process. However, as the estimated Rt drastically decreased following the implementation of Alert Level 4 (Figure 2), I agree that the authors could tell the potential effect of vaccination was not considerable. Furthermore, the authors well discussed the limitation in the revised manuscript. However, if the authors can clearly mention it in the Abstract, that would make the manuscript clearer. I hope this small suggestion makes this manuscript have the highest impact.

Validity of the findings

I was just a little bit confused about the difference between the two figures. I totally agree that showing both figures are very useful.

---

## Round 0.3 · accepted · Accept

The authors should be now congratulated on their successful revisions.